# Low cost audiovisual playback and recording triggered by radio frequency identification using Raspberry Pi

Ádám Z. Lendvai[1,*], Çağlar Akçay[1], Talia Weiss[2], Mark F. Haussmann[3], Ignacio T. Moore[1] and Frances Bonier[4]

[1] Department of Biological Sciences, Virginia Tech, Blacksburg, VA, USA
[2] Department of Biomedical Engineering and Mechanics, Virginia Tech, Blacksburg, VA, USA
[3] Department of Biology, Bucknell University, Lewisburg, PA, USA
[4] Department of Biology, Queen's University, Kingston, ON, Canada
[*] Current affiliation: Department of Evolutionary Zoology and Human Biology, University of Debrecen, Debrecen, Hungary

## ABSTRACT

Playbacks of visual or audio stimuli to wild animals is a widely used experimental tool in behavioral ecology. In many cases, however, playback experiments are constrained by observer limitations such as the time observers can be present, or the accuracy of observation. These problems are particularly apparent when playbacks are triggered by specific events, such as performing a specific behavior, or are targeted to specific individuals. We developed a low-cost automated playback/recording system, using two field-deployable devices: radio-frequency identification (RFID) readers and Raspberry Pi micro-computers. This system detects a specific passive integrated transponder (PIT) tag attached to an individual, and subsequently plays back the stimuli, or records audio or visual information. To demonstrate the utility of this system and to test one of its possible applications, we tagged female and male tree swallows (*Tachycineta bicolor*) from two box-nesting populations with PIT tags and carried out playbacks of nestling begging calls every time focal females entered the nestbox over a six-hour period. We show that the RFID-Raspberry Pi system presents a versatile, low-cost, field-deployable system that can be adapted for many audio and visual playback purposes. In addition, the set-up does not require programming knowledge, and it easily customized to many other applications, depending on the research questions. Here, we discuss the possible applications and limitations of the system. The low cost and the small learning curve of the RFID-Raspberry Pi system provides a powerful new tool to field biologists.

# INTRODUCTION

Using audio or video playback to provide experimental stimuli is a widely used and powerful research tool in behavioral ecology. However, many research questions require that the broadcast or recording of audio or video signals be triggered by a particular event or given to a specific individual within a group or pair. For example, in a study of biparental

Corresponding author
Ádám Z. Lendvai, lendvai@vt.edu

care, *Hinde (2006)* used experimental playback of nestling begging calls directed to only one parent (the male or the female). To achieve this, *Hinde (2006)* set up a portable hide from where she observed the provisioning parents and then launched the playback using a remote control system. Such a system has benefits and limitations. First, while human control provides flexibility in the decision of when the playback should be triggered, it requires the presence of the human observer for each trial, which may not be feasible in many research situations. Second, the presence of the human experimenter also limits the duration, frequency, and sample size of such playbacks. Third, human control is also prone to observer error, limitations on reaction time, limitations on detectability, and potentially observer bias. For example, when the sexes cannot be distinguished unambiguously in the field (as it is the case in many sexually monomorphic species), remote controlling is either impossible or likely involves errors due to misidentification. Finally, for some animals the presence of the human observer could alter behavior and influence responses to experimental stimuli.

Here, we describe an automated event-triggered playback or recording system which couples a passive radio frequency identification (RFID) system with an inexpensive and field-adaptable single board computer, the Raspberry Pi (RPi) (*RPi Foundation, 2015*). RFID systems enable the identification of individuals based on the electromagnetic transmission of a unique radio frequency signal from a passive integrated transponder (a.k.a. a PIT-tag) to a reader (RFID-reader) that detects and stores this information. The small size of PIT tags (currently weighing as little as 0.03 g) and the theoretically unlimited operational lifetime offers a wide range of possible wildlife applications in both invertebrates and vertebrates (*Gibbons & Andrews, 2004*; *Kurth et al., 2007*; *Lauzon-Guay & Scheibling, 2008*; *Bonter & Bridge, 2011*).

The RPi is a credit-card sized computer developed by the Raspberry Pi Foundation. It is a single board computer and thus has all the functionalities of a complete computer, including a microprocessor, memory, and input/output features. In the past decade there has been a growing array of commercially available single-board computers (an up-to-date list can be found online: *Wikipedia, 2015*). RPi currently stands out in this field as the option that is the least expensive and most widely available, and it has a very large user community, making support easily available.

RPi currently comes in different models: model A and model B (with upgrades model A+ and model B+) and the most recent upgrade RPi Generation 2 model B. Model A and A+ are less expensive (US $25 and $20, respectively, through official retailers) and use less power than model B model B+ and Generation 2 model B (all cost $35); therefore, they may be more useful in field conditions where power consumption and research funding are limiting factors. All models are able to run a number of Linux-based operating systems, and have the capabilities of a normal desktop computer. RPi uses micro SD cards to store both the operating system and any data. The memory cards and operating system must be purchased separately (although official retailers offer bundles where the RPi comes with micro SD cards with pre-loaded operating systems).

With the goal to create an easily assembled, inexpensive, and versatile system for multi-functional field use, we coupled the RFID reader developed by *Bridge & Bonter (2011)* with a RPi. Connecting a field-deployable computer to an RFID reader offers a wide range of possible experimental manipulations, where the reader automatically detects specific individuals and the computer analyzes this information and implements specific actions, including playback of audio or visual stimuli (the application described here) as well as providing commands to external devices. In principle, the RFID-RPi system can control any kind of device, including mechanical devices such as food dispensers (for operant conditioning or targeted food supplementation experiments) or traps (for targeted capturing of individuals). Controlling such devices is fairly simple, as the RPi has 17 general purpose in/out (GPIO) pins that can be used to receive and transmit arbitrary binary data. All GPIO functionality is available in a variety of programming and scripting languages, such as Python, allowing for very easy control and access to a wide range of peripheral sensors and devices.

The RFID-RPi system can also be used for event-triggered recording. For instance, video recording is one of the most useful data collection methods in behavioral sciences, but it often generates a lot of 'waste,' i.e. recordings without events of interest. To overcome this problem, a further application of the RPi linked RFID system may be recording of still images or video files only upon detecting target individuals. This function can be achieved either through the Pi Camera modules (either standard or infrared capable modules; they must be purchased separately, cost $25) or through a USB-connected web camera. The same logic can be applied to audio recordings through an external USB-connected sound card with audio recording capabilities.

In this paper, we describe the RFID-RPi set-up and functionalities in detail. Furthermore, in order to test one application of the system under field conditions, we used the set-up to trigger playback of specific audio recordings in the presence of specific PIT-tagged birds. Operating this system does not require advanced programming (such as C or reprogramming the microcontroller of the RFID reader) and can be achieved through simple scripting languages. Although we offer one possible solution and provide the code in Python, different languages can be used to achieve the same goal. Also, although we describe the field test of an audio playback system (because audio playbacks are more widespread in field biology than video playbacks) the following methods can be also applied to implement video playback with little modification (using the RPi's HDMI, composite video or USB connections).

## THE SET-UP

All the elements and their approximate costs are detailed in Supplemental Information 1.

### The RFID reader and antenna

For individual identification, we used a low-cost RFID reader described in *Bridge & Bonter (2011)*. The reader model ("Generation 1 reader") described in *Bridge & Bonter (2011)* can be assembled manually based on the detailed instructions and component list that the authors provide on their website (*Bridge, 2011*). Here, we used "Generation 2" RFID

readers, an upgrade of the published model that can be purchased from Cellular Tracking Technology, Pennsylvania, USA ($35). The Generation 2 readers use SD cards to store the data locally, and provide increased performance and additional options compared with the Generation 1 readers (see details below). The RFID-RPi setup works equally with both models. For this application, a hexagonal or square antenna (diagonally about 6 cm) was connected to the RFID reader that attempted to read a signal for 0.3 s, then paused for 0.2 s to save battery life and then this cycle was repeated continuously. The antenna can be also manufactured easily by hand; instructions are available online (*Bridge, 2011*). The reader records the unique tag number and the current date and time to the seconds in a log file.

## Power source

The RFID reader requires a 12V DC power source. To supply that power, we used small motorcycle batteries (PS1250F1 12 Volt, 5Ah, $8.9 \times 7.1 \times 10.1$ cm, Sealed Lead-Acid Battery, PowerSonic Corp. San Diego, CA) that provide long operation times (several days or weeks, depending on the power saving setups of the RFID reader). The RPi requires a 5 V power source. When electricity is readily available, e.g., in a laboratory, the most feasible option is to power the RPi directly from a power outlet or through the micro USB connector of the RPi. In field conditions, electric power is often not available and a battery is needed. The 12V battery that is used to power the RFID reader can also be used to run the RPi after the voltage has been stepped down to 5 V. To do that, a voltage regulator is recommended, but in our experience the inexpensive USB car chargers also work reliably. In either case, a specific cable needs to be constructed to run both units (RFID and RPi) from the same 12 V power source (Fig. 1). Making this cable is simple: one needs to connect two electric cables into one end of a wire crimp terminal connector and four electric cables into the other end of the connector. It makes a Y-shaped cable, where the basal part of the Y goes to the battery, one branch of the upper part of the Y goes to the RFID reader, and the other branch needs to be soldered to the USB car charger. Evidently, polarity must be respected for all connections; therefore, using colored wires (red and black) is recommended (Fig. 1). The setup then can be insulated with electrical tape.

## The RFID-triggered playback system

Here, we used a model A RPi, but our setup is not model-specific, so our description would work with any RPi model. We used the RS232 serial communication port of the RFID reader to transfer the data to the RPi. While there are RS232 adaptors that are made for the RPi, we instead used a RS232 to USB converter cable (converters using the low-cost PL2303 chipset are automatically detected by Raspbian operating system, so no driver installation is needed, although we experienced variability between the actual cables—see 'Results'). We used a Python (2.7) script on the RPi to process the incoming data and to trigger the playbacks less than a second after the RFID reader sent a PIT-tag detection signal (the actual delay depends on the RPi script settings —see below). Note that Python offers unlimited flexibility to process the incoming information and control the playback to suit specific needs. The operating system can be set up to launch the Python script automatically on startup, so in field conditions, no user interaction is required. This can

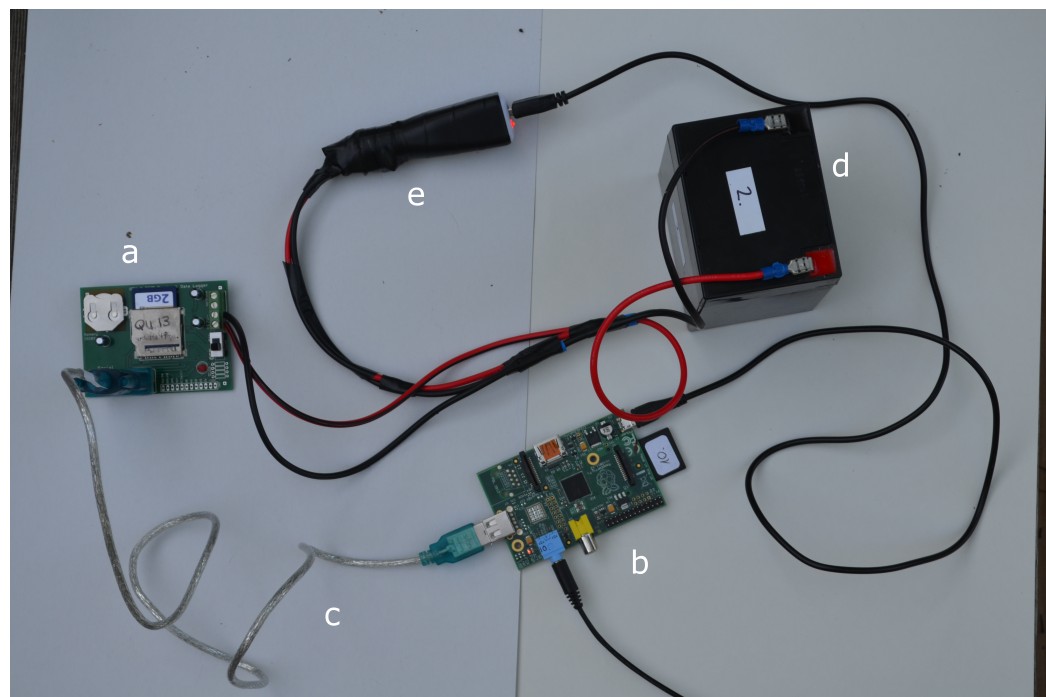

**Figure 1** **The RFID-RPi set-up.** The RFID board (A) is connected to the RPi (B) by a serial-USB cable (C). The battery (D) is connected to both RFID board and the RPi, the latter through a USB car charger (E). Not shown is the RFID antenna.

be achieved in several ways, probably the easiest is to edit the `/etc/rc.local` file (e.g., by typing `sudo nano/etc/rc.local` to the console) and specifying the script, e.g., by `sudo python/home/pi/RFID/rfid.py` & (where the path and the name of the script is followed by &, which specifies that the script should be run in the background).

We provide the Python code Supplemental Information 2 that contains the following functionality. The code (1) triggers the playback of a pre-defined sound file upon reading a target tag and also takes a still picture using the Pi Camera module, and (2) triggers a test playback upon detection of a specified test-tag. This allows researchers to test whether the system is operational in the field. The code also (3) specifies an optional refractory period after a playback, so playback can be prevented during that period (if that period is defined to be greater than 0), and (4) assures that the playback is played only once per detection and is suppressed upon continuous readings (i.e., while the tag remains in the antenna's reading range). Finally, (5) the script creates a log file in the RPi that records the time since startup and the nature of events (target playbacks or test playbacks). Note that in the absence of a display/interface device for the RPi, the script needs to be updated and set-up for playback prior to taking to the field.

The script works with both Generation 1 and 2 RFID readers. The two models use different communication settings (baud rate), and therefore it needs to be adjusted to handle the serial communication correctly. This is done automatically by the script if we supply the reader ID (the Generation 2 readers can be assigned a four character

alphanumeric code, which is recorded in log file) to the script using some naming conventions, see details in Supplemental Information 2.

## Deployment of the playback system in the field

In Spring 2014, we tested and deployed the RFID-RPi playback system in two field sites while conducting a study of parental care in nest-box breeding tree swallows (*Tachycineta bicolor*). All procedures followed guidelines for animal care outlined by ASAB/ABS and the CCAC, and were approved by the Virginia Tech's Institutional Animal Care and Use Committee (#12-020) and the Canadian Wildlife Service (#10771). Briefly, we sought to present female parents but not male parents with playbacks of nestling begging calls every time they entered the box during a specific portion of the period of care for nestlings.

Using nest box traps, we captured male and female tree swallows at two field sites: Davidson College Lake Campus in Cornelius, North Carolina, USA ($n = 45$ females and 43 males) and Queens University Biological Station, Ontario, Canada ($n = 45$ females and 35 males). Each female and male was tagged with a PIT tag integrated with a colored leg band (EM4102 tags from IB Technology, UK). The combined color band/PIT tag weight was 0.1 g (the average $\pm$ SD body mass of tree swallows in our populations is $20.7 \pm 1.2$ for males and $22.9 \pm 1.8$ for females).

The experiment aimed at playing begging call playbacks to half of the females on day 6 of the nestling period (day of hatching = day 0) (the other half of the females were controls). To achieve that, we programmed the RPi by entering the target female's PIT-tag code into the script and uploading the nestling begging stimuli to the memory of the RPi. The programming was carried out the previous day in the lab. For broadcasting the audio stimulus, we used earbud headphones (Sony MDRE9LP) that were attached to the RPi with a 6-foot audio extension cable such that the RFID/RPi set-up could rest on the ground beneath the nest box (Fig. 2). The earbuds played the stimulus files at approximately 55 dB (measured by a portable digital sound level meter) which is comparable to natural begging calls (*Leonard & Horn, 2006*). Nestling begging calls were recorded on nests at the same stage (day 6) by quietly approaching the nestbox and putting a directional microphone into the entrance. The change in the light conditions and the microphone appearing in the entrance elicited begging calls from the chicks. The raw recordings were then edited in Syrinx (John Burt, Seattle, Washington; www.syrinxpc.com) to standardize call rate. We used 10 different stimulus calls that were randomly allocated to the experimental females. Earbuds and audio cables were installed in control nests, but RPi was not connected, and playback was not used.

We deployed the RFID-RPi set-up on day 6. At each deployment, we tested the set-up by passing a test PIT-tag through the RFID reader and listening for playbacks to verify that the system was working. We then installed the earbuds into the nest-box and left the area. After 6 h, we returned to the nest and we tested the playback system again with the test tag and then shut down the system. To shut down the RPi properly, we had to disconnect the serial-USB cable from the RPi that terminated (crashed) the Python process running in the background, because the process expects an active serial connection. Then we attached

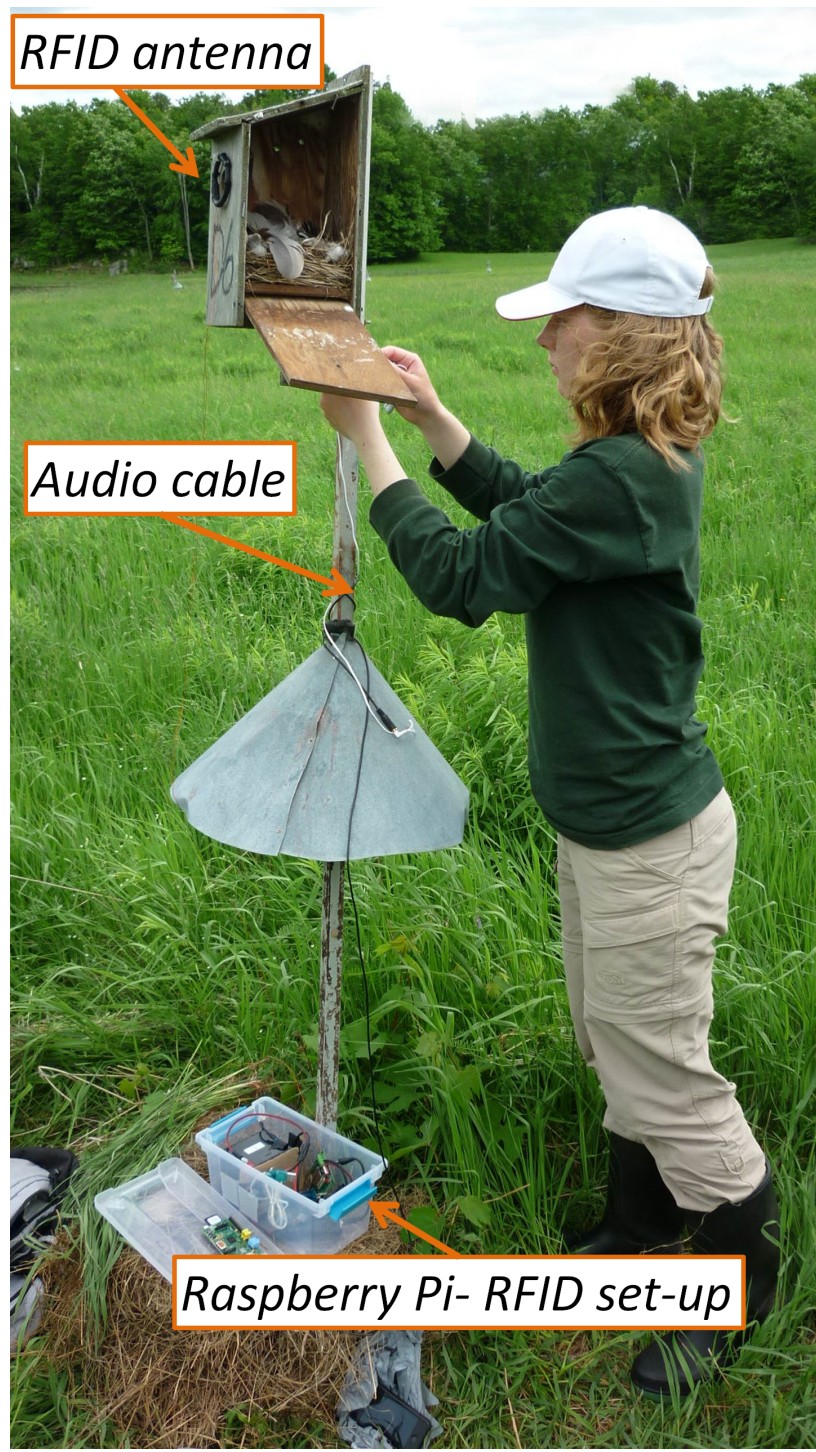

**Figure 2 The field set-up.** The picture shows the RFID antenna fixed to the nestbox entry and the Raspberry Pi-RFID setup in the plastic container (including the battery and the RFID reader). Note that we took this picture when the field setup was being removed; therefore the Raspberry Pi is already removed from its container (and placed on the lid). The white cord around the pole is the cable of the earbud (used to broadcast the playback), and the black cable connected to it (on the predator guard) is the audio extension cable, which is also disconnected from the Raspberry Pi. Photo by AZL.

a keyboard to the USB-port and logged in to the system (by typing a username and password) and then typed `sudo halt` to the command line interface to initiate shutdown. Because we were in the field, we operated the RPi without a display, and so had to type in these commands without being able to monitor the accuracy of our entries. Although it is not recommended, in several cases we simply disconnected the micro USB power source and we never experienced problems associated with the improper shutdown procedure.

## RESULTS

The playback system had a high efficacy. In 21 deployments of the playback system in NC site, the playback system worked in 19 cases. In two cases the playback did not work with the test tag. In both cases, we traced the problem to the Serial-USB Cable (i.e., the RFID readings were not communicated to the RPi), as the same set-up worked with a different cable suggesting that cables should be checked prior to field deployment. In Canada, the playback system worked in 18 out of 20 attempted deployments. In one case, the problem was with the RFID antenna, so the RPi did not receive any signals from the reader. In the other case, the problem could not be identified.

Across all of the nests, the playback system generated an average of $8.89 \pm 0.51$ (SE) playbacks/hr. During this same period, the female visit rate (number of visits per hour, calculated from the RFID records) was $8.96 \pm 0.68$ (SE) per hr, and the two measures were highly correlated across $N = 37$ nests ($r = 0.83, p < 0.6^{-10}$). The visit rate was higher than the playback rate because of the refractory period we specified, so if birds appeared at the box in quick succession (within 2 min from the first entry) then playback was suppressed during this period. Thus, the playback system seemed to work with great efficiency.

### Limitations

The RPi is a versatile and inexpensive system that promises to be valuable in field studies that call for automated data collection or playback. However, there are some drawbacks to the RFID-RPi system. The first drawback is the need for a display and interface device (e.g., keyboard) for troubleshooting the RPi in the field. In conditions where the field site is not far away from a home base, or where the RPi is used in the lab, this drawback may not be significant. However, troubleshooting problems with the RPi in remote field locations can be a significant hurdle. This drawback can be overcome with the use of one of several custom made display/interface devices available on the market that can connect to a RPi. Smaller displays (TFT board monitors) can be connected to the GPIO pins of the RPi, and do not require an independent power source, but their small size may be a limitation in certain cases. Alternatively, external monitors can be attached through the HDMI port, but these may require their own power source that (depending on the device) may have a different voltage requirement than the RPi or the RFID reader. Finally, an external Bluetooth transceiver can be connected to a USB port and it could be used for wireless data transfer.

A second drawback is the lack of a real-time clock (RTC) in the RPi. This can be circumvented by synchronizing the RPi with an external device that does have a real-time clock (such as the RFID reader). For example, in our playback system, we used the test tags

in the first deployment to create an entry in both the RFID log with a timestamp and in the Python script log. Since our Python script recorded the time elapsed between recorded events, the real time entry of the first detection of the test tag could be used to assign real times to all subsequent entries in the RPi log. Additionally, there are ways to install an external RTC with dedicated battery to the RPi. Alternatively, the internal clock of the RPi can be updated upon startup—but again, this requires a display unless these operations can be performed blind.

Although it may not be a limitation in most cases, one should be aware that the RPi, being an exposed chipboard, needs good protection from the elements in field conditions. There are commercially available cases for the RPi that are waterproof; however, when the required number of systems is large, these cases may be prohibitively expensive. In our deployment, we used a plastic container (roughly the size of a shoe-box) that held the RFID board, RPi, the batteries, and all the cables (Fig. 2). The plastic containers were waterproof when sealed. In such a system boards should not touch each other or the battery, which can be avoided by using a a divider made from a non-conductive material, e.g., plastic. Another important factor to be considered is the overheating of the system, especially in open field condition with direct sun exposure. In such cases, the set-up needs to be shielded by a cover to prevent over-heating.

An additional limitation comes from the RFID reader itself: the reader cannot detect more than one tag at a time, and when multiple tags are present in the antenna's reading range, the device is not capable of storing any tag's record (*Bridge & Bonter, 2011*). By reducing the size of the antenna however, one can minimize this effect. For instance, in our study, it is possible that in some cases both the males and the females were present at the nestbox entrance at the same time, and we missed those readings. However, because only one bird can pass through the entrance at a given time, this effect is only momentary and unlikely to bias the results. However, if the tags are not present at the antenna's reading range simultaneously but in quick succession, the setup can deal with those cases. The RPi may be programmed to withhold playback in these cases when two tags are registered in close temporal proximity

## CONCLUSION

We present here a flexible, versatile, and relatively inexpensive system for automated data-logging and audio or video playback and recording in field conditions. We believe the RFID technology, already widely used in ecology and evolution, when coupled with the RPi or other single board inexpensive computers, presents many interesting opportunities for experimental and observational field research.

## ACKNOWLEDGEMENTS

We thank Eli Bridge for help and troubleshooting with the RFID readers. We thank Miyako Warrington and David Wilson for useful and constructive comments on an earlier version of this manuscript.

### Funding

Funding was provided by a U.S. National Science Foundation (NSF) grant (Frances Bonier, Ignacio T. Moore, and Mark F. Haussmann; IOS-1145625), the Natural Sciences and Engineering Research Council of Canada, and a Banting Postdoctoral Fellowship (FB). During the preparation of the manuscript, Ádám Z. Lendvai was supported by a Hungarian Scientific Research Fund (OTKA K113108). The funders had no role in study design, data collection and analysis, decision to publish, or preparation of the manuscript.

### Grant Disclosures

The following grant information was disclosed by the authors:
U.S. National Science Foundation (NSF): IOS-1145625.
Natural Sciences and Engineering Research Council of Canada.
Banting Postdoctoral Fellowship.
Hungarian Scientific Research Fund: OTKA K113108.

### Competing Interests

The authors declare there are no competing interests.

### Author Contributions

- Ádám Z. Lendvai and Çağlar Akçay conceived and designed the experiments, performed the experiments, analyzed the data, wrote the paper, prepared figures and/or tables, reviewed drafts of the paper.
- Talia Weiss conceived and designed the experiments, reviewed drafts of the paper.
- Mark F. Haussmann, Ignacio T. Moore and Frances Bonier contributed reagents/materials/analysis tools, reviewed drafts of the paper.

### Animal Ethics

The following information was supplied relating to ethical approvals (i.e., approving body and any reference numbers):

All procedures followed guidelines for animal care outlined by ASAB/ABS and the CCAC, and were approved by the Virginia Tech's Institutional Animal Care and Use Committee (#12-020) and the Canadian Wildlife Service (#10771).

### Supplemental Information

Supplemental information for this article can be found online at http://dx.doi.org/10.7717/peerj.877#supplemental-information.

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
