# Peer review of "Low cost audiovisual playback and recording triggered by radio frequency identification using Raspberry Pi"

_PeerJ, doi:10.7717/peerj.877_

## Round 0.1 · original submission · Minor Revisions

Dear authors,

Both of the reviewers were positive about your manuscript and I would like to encourage you to make the revisions suggested in order to improve your manuscript so that it is suitable for publication in PeerJ.

In general I found both reviewers to be very constructive in their comments and from my reading, you should be able to affect changes to most of them relatively easily. Reviewer 2 was not so explicit about the required changes made, but I do agree with the sentiment expressed here that most biologists are generally not as technically minded as perhaps yourselves. In areas your manuscript could be a little more clear about technical details, to guide and help non experts through more, and this was a point also picked up by Reviewer 1.

I myself wondered what a Raspberry Pi was exactly, and I do think it might be useful to have a brief discussion about that and the different models available (see reviewer 1) and the possibility that this could be substituted for any other microcomputing hardware (see reviewer 2). There is an obvious issue here that if your set-up is too specific to a particular model of Raspberry Pi, then what will happen in a few years if that model is no longer available? Some more detail here to cover all these points will help to broaden and lengthen the applicability of your paper. I also think it might be useful to consider discussing the issue raised by reviewer 1 with respect to interfacing with the Raspberry, as again that might make your set-up a little bit more user friendly.

I agree with the first two substantive points raised by Reviewer 1, and they certainly need to be dealt with (with respect to point 1 I agree that their second option will be best as long as you are confident that the video playback options will work as well as the audio playback options). I do also encourage you to try and deal with all of the minor comments that both reviewers raised as well.

Note that the Comments from the second reviewer are listed as being an annotated MS, but in fact it is just a PDF that coments the reviewers comments.

·

Basic reporting

no comments

Experimental design

no comments

Validity of the findings

no comments

Additional comments

Review of "Low cost audiovisual playback and recording triggered by radio frequency identification using Raspberry Pi"

This methodological paper describes an innovative application of two existing technologies. The first is a radio frequency identification system that is used to detect and identify individual animals that have been fitted with unique RFID tags. Since the tags weigh less than half a gram, they can be attached to a diverse range of animal taxa. The second technology is a small computer known as a Raspberry Pi. When a radio frequency identification tag is detected, its unique code is sent to the computer, which is programmed to control one or more of a wide range of peripheral attachments. In the current study, the attachment was a miniature loudspeaker that broadcasted begging vocalizations whenever the mother bird approached its nest.

When combined, these two technologies provide potentially unlimited opportunities for conducting automated, individual-activated, individual-directed manipulations in the lab or field. For example, it can be used to broadcast audio or video stimuli, as was done here, to record specific events using photos or audio/video recordings, or to control motors or solenoids that operate food dispensers, gates, or other devices. The overall system is simple and appears to be reliable under field conditions. It is also highly affordable, with an overall system appearing to cost less than $100 USD. Given these characteristics, I am confident that this technology will be adopted by a wide range of biologists working on a wide range of taxa. I have a few minor suggestions to help improve this manuscript.

(1) The paper describes a system that records and that conducts audio and video playbacks (see, for example, the title and the first sentences of the Abstract and Introduction). Yet this description does not match what was done in the study, or what could be accomplished by the system. One option would be to limit the described applications of the system to acoustic playback, since that was the only application that was actually tested in the current study. The authors could then mention other possible applications in the discussion. A second option would be to describe the system in more general terms. For example, a field-deployable system that automatically detects specific individuals, and which implements experimental manipulations to those individuals. The paper could then describe the range of possible manipulations, and then describe a field test of one possible manipulation based on acoustic playback. I recommend the second option because it highlights the system's broad potential application.

(2) The description of the RFID system is offloaded to a reference to Bridge & Bonter 2011 (line 50). However, since it is an integral component of the overall system, it should be described in some detail here. What are the model numbers of each component, what were their specifications, what was their cost? The cost is important since the overall system is described as "affordable". In the Bridge & Bonter paper, the RFID system costs less than $40, but this should be updated for the current study. Another particularly important detail is whether or not multiple tags can be detected simultaneously by the RFID reader. Bridge & Bonter 2011 state that the system "cannot read tags when more than one is present in the read range." Therefore, in the current study, females may not have been detected if they approached the nest box while accompanied by their mate. This might explain, in part, why there was not a better correlation between the number of visits and the number of playbacks. Other applications would also require an ability to detect multiple tags simultaneously. For example, I can imagine several scenarios where the system activates a device when it detects a specific individual (e.g., a playback of the individual's mate's song), but only if another individual (e.g., the mate) is not present. These capabilities/limitations should be discussed.

(3) More details are also needed about the Raspberry Pi. Which model was used (the paper mentions 3 models, but does not state which was used), what did it cost, how are stimuli and recorded data stored (I assume on a micro SD card; if so, what formats and capacities are available and compatible with the computer?), is the Pi camera module built into the device or attached to one of its ports, what software is needed to record photos, sound, and video? If the RFID system can detect multiple tags simultaneously, then can the computer apply contingencies such as "broadcast sound when animal X is detected, but only if animal Y is not also present"?

(4) On a related note, it would also be useful to know the approximate price of one entire system, including cables, battery, RFID reader, and computer. Obviously, prices vary, but approximate costs could be provided with appropriate caveats. I think the system's affordability is one of its most attractive features, so approximate costs should be provided wherever possible.

(5) Lines 135 - 139: Unless I missed it, "female visit rate" and "feeding rate" were not mentioned in the methods. How were these variables defined, and how were they quantified? Were they quantified by a human observer? Also, if the RFID system cannot detect tags when more than one is present, then this could account, in part, for the imperfect correlation reported here.

(6) Lines 181 – 190: if known, please specify the range of operating conditions (e.g. temperature, humidity) for each component and for the overall system. Also, report the range of conditions from the current study (inside and outside the plastic box, if available), since these were clearly within the acceptable range.

(7) Figure 1: I recommend making this a 2-panel figure. In the first panel, include the current photo, but perhaps expand it to include the playback headphones and the RFID antenna. In the second panel, show a photo of a field-ready system that is enclosed in a weatherproof box (perhaps with the lid off).

Specific suggestions:
(8) Line 30 - change to "distinguished unambiguously"

(9) Line 66 - awkward sentence. Please revise.

(10) Line 88 - please clarify whether the script also records the RFID tag ID

(11) Lines 126 – 131 - please explain why there were fewer deployments (N=41) than tagged females (N=90). I assume that a system was set up only for the experimental females that received playback, and not for the control females that did not?

(12) Lines 202 – 205 - please remove the all-caps from the titles of these two articles.

I enjoyed reading this paper.

Sincerely,
Dave Wilson

·

Basic reporting

No comments

Experimental design

No comments

Validity of the findings

No comments

Additional comments

See attached comments

---

## Round 0.2 · Minor Revisions

I am happy to accept this revised version for publication and I think you have done a good job with your revisions.

There is just one awkward part of the manuscript that needs addressing. On line 29 you have written:
"To achieve this, she set up a portable hide from where she observed the provisioning parents and then launched the playback using a remote control system."

This sentence really needs to be referred back to the discussed study so I suggest changing to

"To achieve this, Hinde (2006) set up a portable hide from where she observed the provisioning parents and then launched the playback using a remote control system."

Thanks

---

## Round 0.3 · accepted · Accept

Thanks for making that small change. Apologies for the delay on my part, I was away in the field.